# Planning with emission models reduces the carbon footprint of new reservoirs
Tomasz Janus [1] ✉, Christopher Barry [2], Shelly Win [3] & Jaise Kuriakose [1] ✉

Reservoirs collectively contribute 1–2% of global anthropogenic greenhouse gas emissions, although individual emissions can vary widely. While emission models have considerably advanced our understanding of the lifetime carbon impacts of reservoirs globally and offer means to inform judicious planning, their widespread adoption is hindered by high manual processing requirements, uncertainties, and linkages to geospatial drivers that can be obscure for planners. Meanwhile, simpler Tier 1 methods fail to capture variability across individual reservoirs and can overestimate national emissions by 50% compared to model-based estimates. Here we introduce an automated and transparent framework for large scale reservoir emission assessments and planning with spatially-explicit emission models to address key limitations in current approaches. By applying our framework to strategic hydropower expansion in Myanmar, we show how emission models can support low-carbon reservoir development at large scales. Our results show that the proposed methodology can yield a hydropower strategy for Myanmar that eliminates 0.94 $MtCO_{2e}$ in emissions (1% of national total), conserves 239 $km^2$ of forest and arable land, and reduces the number of barriers in lower river reaches from 28 to 7.

Reservoirs can generate substantial greenhouse gas (GHG) emissions, predominantly of $CO_2$ and $CH_4$[1], by creating environments conducive to organic matter synthesis and degradation[2,3] altering the carbon cycle of the natural landscape. The primary cause of emissions is the decomposition of organic matter flooded after dam construction, transferred to the reservoir by runoff, or produced in the reservoir, e.g., from algal growth[3]. Recent estimates indicate that globally reservoirs contribute approximately 5.2% of total anthropogenic $CH_4$ emissions and 0.2% of $CO_2$ emissions; equivalent to 1–2% of total anthropogenic GHG emissions[4]. Given that reservoirs are important contributors to global emissions, it is crucial to accurately assess the carbon footprint of both existing and planned reservoirs. This need is particularly pressing amid ongoing investments in hydropower, with approximately 3700 hydroelectric dams – each exceeding 1 MW in capacity —either under construction or planned[5,6].

Emissions from individual reservoirs vary widely, with areal fluxes differing by up to two orders of magnitude[7], driven by location-specific geoclimatic and topographic factors. Reservoirs in low-lying tropical regions are among the largest contributors to water body GHG emissions globally[8–11], though similarly high emissions have also been discovered in some temperate mid-latitude reservoirs[12,13]. This variability highlights the need for examining reservoir expansion strategies that minimize carbon footprints. However, incorporating emissions as a criterion in decision-making, such as in strategic dam planning frameworks, poses two key challenges: i) Comprehensive models like G-res[14] require extensive input data, limiting their applicability to large-scale national and regional reservoir inventories. ii) The complexity of these models can make them opaque and difficult to integrate as practical decision-making tools.

Recent studies on hydropower planning have begun incorporating GHG emissions as a criterion for dam selection[10,15,16]. However, due to the lack of methodologies for estimating emissions from large reservoir inventories using emission models, these studies instead relied on IPCC Tier 1 emission factors (EFs)[17]. This approach assumes that areal fluxes for a given gas or emission pathway remain constant across all reservoirs sharing the same dominant explanatory variable, such as latitude[11] or climatic zone[4]. While this method allows estimating emissions with minimal effort and has been essential for global reservoir emission assessments[4,8,9,11], it was also found to introduce large biases and inaccuracies[7]. For this reason it is essential to move beyond emission factors by embracing frameworks that leverage emission models for the planning of low carbon reservoirs and assessing their impacts on global emissions.

To address this need, we automate input data collection for the G-res model[14], enabling its integration into large-scale planning of low-carbon reservoirs and GHG emission assessments. This step is designed to be generic, allowing integration with spatially explicit emission models that

[1]Tyndall Centre Manchester, University of Manchester, Manchester, UK. [2]UK Centre for Ecology & Hydrology, Bangor, UK. [3]International Water Management Institute (IWMI), Myanmar Office, Yangon, Myanmar. ✉e-mail: tomasz.janus@manchester.ac.uk; jaise.kuriakose@manchester.ac.uk

may be developed in the future. By accounting for local and catchment-scale biogeochemical and physical characteristics, such models can estimate emission differences between individual reservoirs, allowing for more precise selection of low-carbon options compared to the Tier 1 emission factor-based approaches. We further enhance our framework with explainable artificial-intelligence (xAI)[18], a set of methods related to sensitivity analysis (SA)[19] that emphasize human-centered visualization and interpretability. By doing so, we increase model transparency and enable visual interpretations of model results. This step further supports decision-making, where information is critical, particularly in planning large infrastructure projects like reservoirs, where choices are often irreversible and have far-reaching systemic consequences[20].

By applying our framework to reservoir emission assessment and strategic low carbon dam planning in Myanmar, we investigate three key research questions. First, how do G-res model outputs compare to estimates derived from Tier 1 emission factors for individual reservoirs and nationally? Second, how do these differences influence asset selection in strategic low carbon dam planning? Finally, what additional insights can emission models provide beyond emission estimates, and how can this information enhance decision-making and contribute to a detailed understanding of the sources and patterns of reservoir emissions at regional and national levels?

Our study reveals substantial differences in emission estimates between G-res and Tier 1 emission factors – over 200% for individual reservoirs and around 50% nationally. These differences can substantially impact optimal dam configurations in strategic low carbon planning, highlighting the importance of a wider adoption of spatially explicit emission models. By leveraging spatial metrics natively supported by our framework alongside emissions, we found low-carbon hydropower portfolios with emission intensities comparable to wind and solar while minimizing land loss and, implicitly, reducing assets in lower river reaches.

Although our analysis focuses on Myanmar, our methodology addresses the broader challenge of mitigating emissions from planned reservoirs worldwide. We estimate that the 3700 planned hydroelectric dams, expected to add approximately 720 GW of capacity over the next 10–20 years[5], could contribute about 400 $MtCO_{2e}$ annually – equivalent to 11% of the EU's emissions in 2022. Integrating emission models into planning presents a rare opportunity to tangibly restrict reservoir emissions globally.

## Advancing emission modeling with automation and transparency

Our framework, illustrated in Fig. 1, combines a spatially-explicit GHG emission model with an automated system for sourcing input data from geospatial datasets and enhances it with machine learning (ML) and xAI. Automation enables estimating emissions from large numbers of reservoirs in a short amount of time, with minimal manual processing, which is a key enabler for adopting emission models in large-scale assessments and planning. In the first step, we calculate the contours of reservoirs and catchments for a suite of selected dams within a given geographical extent. We then retrieve data from multiple open global spatial datasets and use geometric operations and statistical analysis to extract key hydro-geomorphological, biogeochemical, and climatic parameters for reservoirs and their catchments, which drive emission predictions.

To enable a deeper understanding of the model and its outputs, we first develop a surrogate model that replicates the behavior of the original emission model. This surrogate model, thanks to its special structure supporting interpretations, is used to explain the original model through a suite of xAI methods that visualize and explain model predictions both globally (model-level) and for each individual reservoir (instance-level). This information aids in understanding the model's behavior and helps to

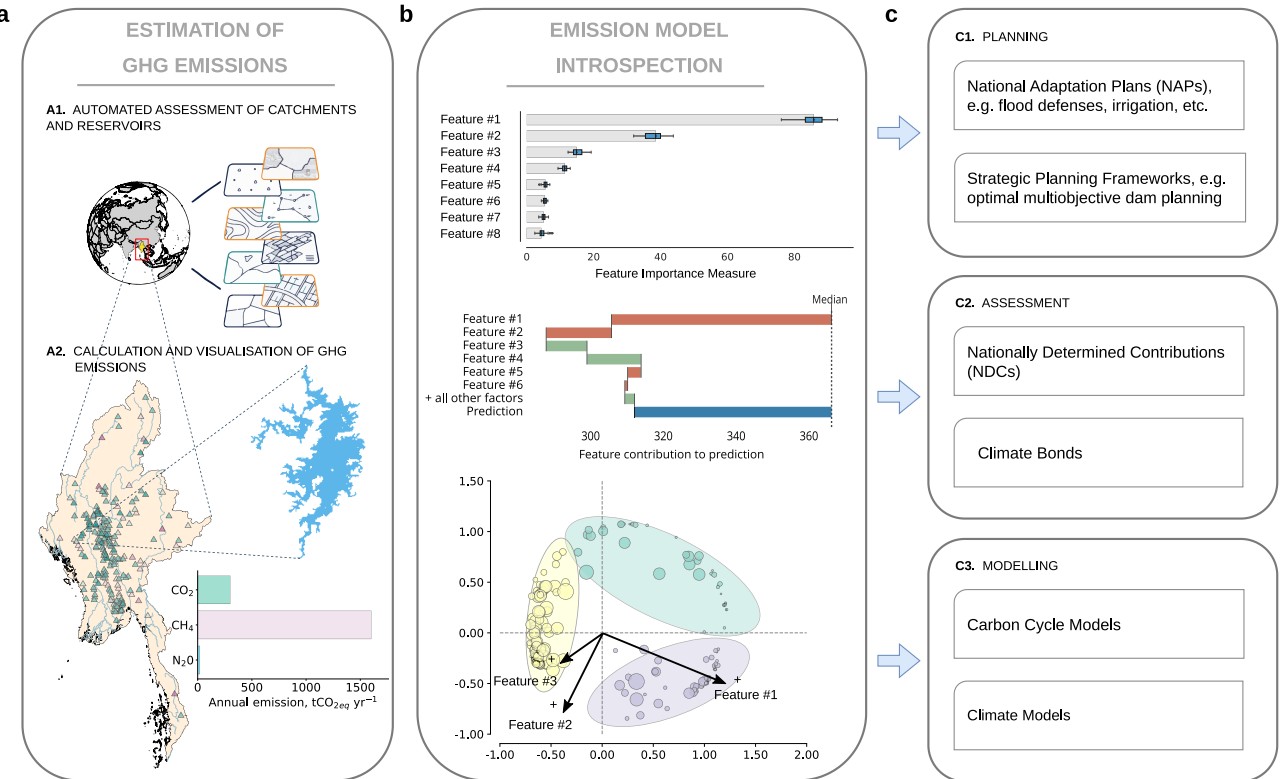

**Fig. 1 | An automated and transparent framework for integrating spatially-explicit emission models into reservoir assessments, investment planning and larger modeling frameworks. a** Calculation of emissions happens in two steps - (A1) automated collection of GHG emission model inputs from global spatial datasets and (A2) emission calculations using the collected input data. **b** We apply machine learning (ML) and xAI to analyze and explain the behavior of the emission model tailored to improve understanding of key factors driving emission predictions. **c** The outputs generated by the framework can support emission assessments, provide inputs to other modeling frameworks, and aid in developing policies aimed at transitioning to low-carbon futures in sectors such as energy and agriculture.

validate its outputs against known outcomes and expert knowledge. Although relatively new to environmental applications, xAI has already shown measurable success in high-stake decision-making in healthcare[21] and finance[22].

Beyond planning, the framework can be applied to assess current reservoir emissions for inclusion in national GHG inventories and Nationally Determined Contributions (NDCs). It can also be used to produce Tier 2 and Tier 3 emission estimates, in line with the IPCC's recommendation to use the G-res model where sufficient data is available[17]. It is supported by two free open-source packages - RE-Emission[23] and GeoCARET[24]. Further technical details are provided in the Methods.

## National-scale case study for Myanmar

We use Myanmar as a case study to demonstrate the utility of our framework in assessing the total carbon budget of reservoirs at a national scale and in planning country-wide low carbon reservoir investments.

Myanmar is an ideal location for estimating reservoir emissions and testing emission models due to its diverse geography and climate, which encompass broad gradients of parameters that influence emission rates. Myanmar's climate is regional, with the coastal regions and northern highlands receiving high and moderate rainfall, along with moderate temperatures. In contrast, the central dry zone experiences relatively low rainfall of <1000 mm annually and extreme summer temperatures often exceeding 40 °C. This geographic and climatic diversity supports a variety of biomes, ranging from tropical rainforests to dry forests and mangroves. Additionally, the country's uneven population distribution, with higher densities in the central lowlands that support intensive agriculture, contributes to varying levels of anthropogenic impacts, notably nutrient runoff, which influence reservoir emissions. The diverse topography from lowland plains to elevated highlands, leads to wide variation in reservoir morphologies: shallow reservoirs are prevalent in the lowlands, and deeper reservoirs in the highlands.

Myanmar is also investing in the hydroelectric sector, and should enhance the resilience of its water resources to climate change by improved retention and storage. The country has over 150 irrigation reservoirs and over 70 planned hydroelectric projects, totaling approximately 25,865 MW of generation capacity, with only 3215 MW currently installed. Despite uncertainty following the 2021 military coup, Myanmar remains committed to expanding its hydropower development[25]. However, many of the planned hydropower projects are concentrated in the ecologically sensitive Irrawaddy and Salween rivers, two of the world's few remaining free-flowing rivers[26]. Given the scale of planned investments and the potential negative impacts of reservoirs, including GHG emissions, particularly in eco-sensitive regions[27,28], Myanmar needs a strategic study for hydroelectric dam planning that incorporates GHG emissions as a decision criterion. Moreover, recent droughts exacerbated by climate change in the central dry zone necessitate a reevaluation of agricultural water storage and its carbon footprint.

## Results

### Reservoir emissions in Myanmar

We quantified GHG emissions from Myanmar's reservoirs using both the G-res model and global Tier 1 climate-zone-based emission factors[4]. Here, we highlight the novel capability of our framework to conduct nationwide assessments of reservoir emissions with models that incorporate location-specific emission drivers. We compare the estimates obtained from both methods on the scale of individual reservoirs and nationwide, and use predictions from G-res to derive country-specific emission factors.

We show that net biogenic areal emissions vary widely among reservoirs, ranging from <500 $gCO_{2e}/m^2/yr$ to >5000 $gCO_{2e}/m^2/yr$ (Fig. 2). Emissions are highest and most variable among irrigation reservoirs, which tend to have smaller surface areas and are concentrated in the central, low-altitude, and warm region of the country. Hydroelectric and multipurpose dams, typically located further upstream in river networks, are larger and have generally lower areal emissions, though some exceed 2000 $gCO_{2e}/m^2/$

yr (Fig. 2b, c). Substantial differences in predicted areal emissions can be observed between reservoirs in close geographical proximity, particularly at lower altitudes, where geomorphological differences in reservoirs and their catchments are more pronounced. In these flatter landscapes, variations in littoral zone size can considerably influence methane emissions, leading to large disparities in areal emission rates.

We observe discrepancies of up to and exceeding 2000 $gCO_{2e}/m^2/yr$ between predictions from G-res and global Tier 1 emission factors. These differences are most pronounced in irrigation reservoirs, though substantial discrepancies also occur among hydroelectric and multipurpose reservoirs (Fig. 2b, c). Notably, G-res estimates tend to exceed those from Tier 1 emission factors at the higher end of the emission spectrum, while the opposite is true for lower emission ranges. This highlights the limitations of Tier 1 emission factors in capturing the variability in emissions that depend on a range of spatial and environmental drivers that can vary widely between individual reservoirs. The relative differences in predicted areal emissions between the two methods can exceed 200% for both net and gross emissions and surpass 700% for specific emission pathways, such as $CH_4$ ebullition from hydroelectric reservoirs (Supplementary Figs. 6 and 7).

We also quantified gross emission factors for four pathways: $CO_2$ diffusion and $CH_4$ diffusion, ebullition and degassing, to establish Myanmar-specific Tier 1 emissions, using G-res predictions for upscaling. Given the higher uncertainty in emission estimates from irrigation reservoirs, owing to G-res regressions being derived primarily from hydroelectric measurements, possible drying-rewetting cycles, and limited operational data, we report recalculated emission factors from G-res for all reservoirs combined and, additionally, for hydroelectric reservoirs only. The results presented in Table 1 reveal differences between the global and calibrated emission factors, particularly for $CH_4$ ebullition and degassing in warm temperate dry climates, as well as for other $CH_4$ pathways across various climate zones. These findings suggest that region- and country-specific emission predictors should be established whenever possible, as global emission factors may be biased toward reservoirs from other geographical contexts, leading to imprecise estimates both individually and in total.

Our study estimates that Myanmar's existing reservoirs emit 3.84 $MtCO_{2e}/yr$, comprising 2.34 $MtCO_{2e}/yr$ from 152 irrigation reservoirs, 0.29 $MtCO_{2e}/yr$ from 5 hydroelectric reservoirs, and 1.21 $MtCO_{2e}/yr$ from 13 multipurpose reservoirs. By comparison, estimates based on global emission factors are 17% higher overall (4.45 $MtCO_{2e}/yr$) and 86% higher for hydroelectric reservoirs (0.54 $MtCO_{2e}/yr$). These estimates adopt a deliberately inclusive approach, as they account for $CH_4$ degassing emissions from irrigation reservoirs. Although such emissions are unlikely in these systems, we retain them here to acknowledge the possibility that irrigation reservoirs may emit more than typically estimated[29,30], the substantial uncertainty associated with these systems, and the probable omission of numerous smaller reservoirs at the national scale. Excluding degassing emissions from irrigation reservoirs lowers the total national-scale emissions to 2.62 $MtCO_{2e}/yr$ based on G-res estimates and 3.57 $MtCO_{2e}/yr$ based on global emission factors. Projected emissions from 40 planned hydroelectric and multipurpose reservoirs are 3.55 $MtCO_{2e}/yr$, while Tier 1-based calculations yield values 57% higher, at 5.59 $MtCO_{2e}/yr$.

Further details on reservoir emissions in Myanmar are provided in Supplementary Tables 1 and 2, and – specifically for hydroelectric units – in Supplementary Fig. 2. The differences between emissions predicted by our framework and those derived from climate-zone-based emission factors are further examined in Supplementary Results 1.2.2 and visualized in Supplementary Figs. 4, 5, 6, and 7.

### Supporting model outputs with explanations

**Instance-level explanations.** We selected two planned hydroelectric reservoirs, Belin and Laza, to illustrate how instance-level explanations can reveal the key sources of emissions and their variations across sites. We visualized these differences in Fig. 3 using breakdown plots[31] that decompose model predictions into sums of contributions from individual input variables. The green and red bars represent, respectively,

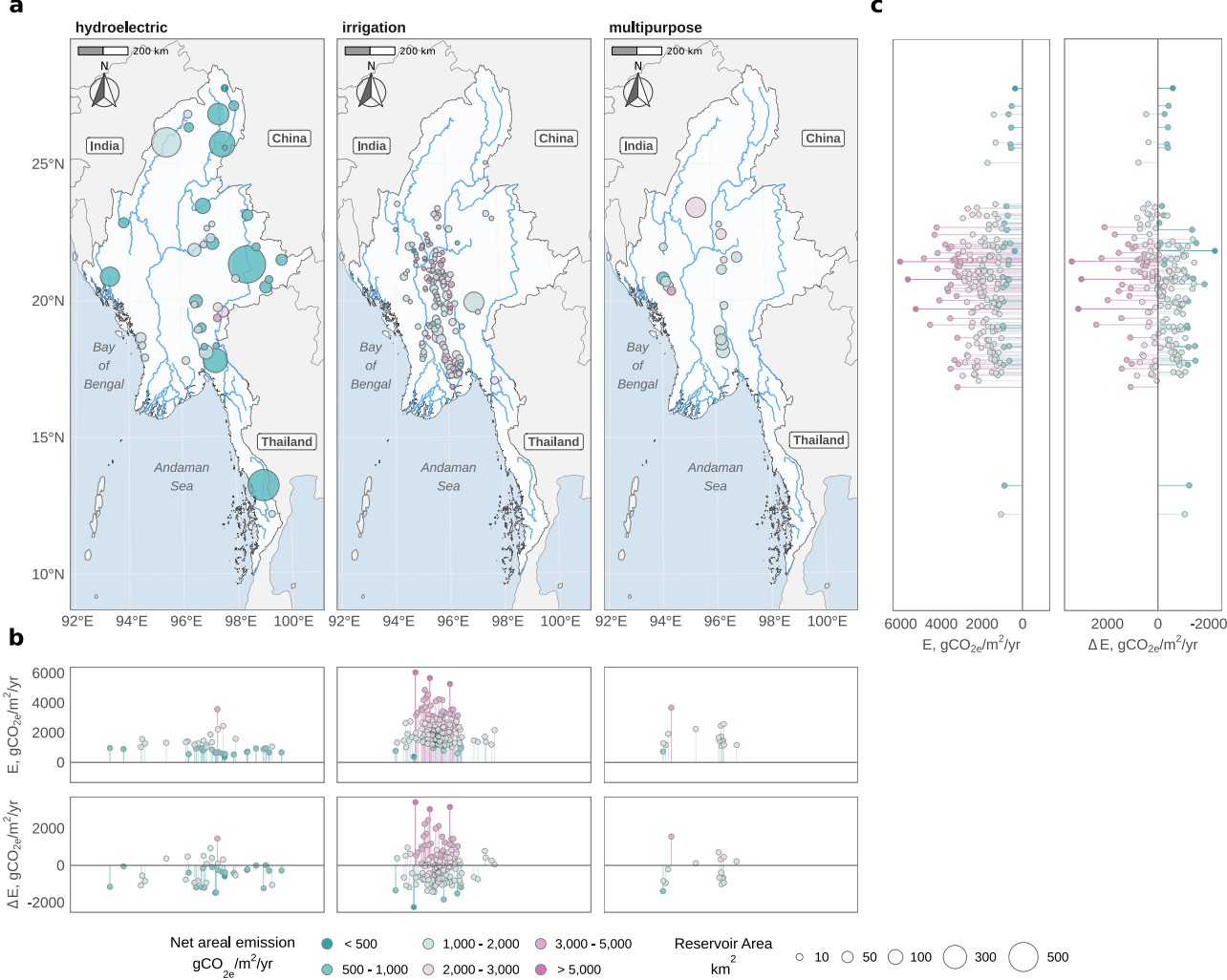

**Fig. 2 | Reservoir emissions and their variability by reservoir type and location in Myanmar. a** Reservoir markers are scaled by surface area and colored by anthropogenic (net) areal emissions. **b, c** Net areal emission projection from G-res ($E$) and differences between net areal emissions from G-res ($E_{G\text{-}res}$) and global Tier 1 emission factors ($E_{Tier1}$)[4]. $\Delta E = E_{G\text{-}res} - E_{Tier1}$.

positive and negative attributions relative to the mean prediction across all reservoirs (the intercept), while the blue bar indicates the difference between the individual prediction and the mean prediction. A complete list of input features used in this analysis is provided in Supplementary Table 5.

We show that although Belin and Laza have similar projected net areal and total emissions overall, their respective areal $CO_2$ and $CH_4$ emissions are different and explained by different attributions of inputs. Belin's net areal $CO_2$ emissions (253 $gCO_{2e}/m^2/yr$) fall below the national mean (366 $gCO_{2e}/m^2/yr$) because of a low forested area fraction (0.12), high shrub cover (0.783), and above-average runoff and catchment slope. In simple terms, Belin has below-average $CO_2$ emissions because it inundates a smaller forested area compared to the average reservoir. Forests located on mineral soils in tropical act as carbon sinks, but inundation eliminates this capacity. The resulting loss of the sink function is therefore accounted for as a positive (source) flux. Consequently, constructing reservoirs on forested land leads to substantial net $CO_2$ emissions. Moreover, the reservoir's short hydraulic residence time, driven by high runoff, further limits the conversion of organic matter into $CO_2$.

In contrast, Laza inundates an area with a high forested fraction (0.755). In isolation, this would elevate its net $CO_2$ emissions by approximately 148 $gCO_{2e}/m^2/yr$ above the national average and to 514 $gCO_{2e}/m^2/yr$ overall. However, due to the reservoir's high altitude, the below-average evapotranspiration and air temperatures contribute to the reduction of the

predicted $CO_2$ emissions by 159.4 $gCO_{2e}/m^2/yr$ (111.1 $gCO_{2e}/m^2/yr$ due to low evapotranspiration and 48.3 $gCO_{2e}/m^2/yr$ due to low air temperature). The substantial additional contributions from other inputs (denoted as "+ all other factors" in Fig. 3), underscore the complex interplay of multiple drivers influencing $CO_2$ emissions.

The estimated net areal $CH_4$ emissions for both reservoirs are below average, primarily due to above-average depths, which suppress methane production. Furthermore, their large volumes are identified as factors contributing to lower $CH_4$ emissions. This is because shallow irrigation reservoirs in Myanmar's warm, dry regions – hotspots for methane emissions – tend to have small volumes. As a result, methane emissions are inversely correlated with both mean depth and volume. Contrary to Belin, Laza has a large catchment area which is predicted to add 1359 $gCO_{2e}/m^2/yr$ of $CH_4$ emissions above the national average. This may be due to greater inputs of organic material and nutrients from larger catchments, which stimulate productivity and methanogenesis. However, its large depth and volume, along with below-average air temperature, strongly reduce the final prediction to 226 $gCO_{2e}/m^2/yr$, which is considerably lower than the national mean.

**Model-level explanations**

We processed and visualized instance-level additive feature attributions across all reservoirs, thereby elucidating patterns in emission drivers nationally. Figure 4a, b shows the distribution of reservoirs in a coordinate

https://doi.org/10.1038/s43247-025-02899-6 **Article**

space defined by additive feature attributions, calculated using the breakdown method[31], and reduced to two dimensions via Principal Component Analysis (PCA). The reservoirs were classified into categories based on their dominant $CO_2$ and $CH_4$ emission drivers, referred to as C-Categories and M-Categories, respectively, and visualized on maps using Voronoi cells.

We observe that emission drivers are strongly linked to geographical location; Fig. 4c, d, although high variability in emissions is observed within the categories, as reported earlier. The central dry zone is occupied by reservoirs belonging to C-Category 1 and M-Category 1 – having large $CH_4$ emissions due to low mean depths with greater littoral area proportions and high and medium-high ambient temperatures, and average $CO_2$ emissions promoted by temperature but also moderated by relatively small proportion of forest in the impounded areas. These reservoirs are predominantly used for irrigation and have high or very high net areal emissions between 1000 $gCO_{2e}/m^2/yr$ and $> 5000$ $gCO_{2e}/m^2/yr$.

Reservoirs in C-Category 0 are situated at higher altitudes, where lower temperatures limit $CO_2$ emission rates. However, some of the reservoirs are located in forested areas, contributing to high net $CO_2$ emissions via loss of former landscape C sinks with inundation. C-Category 2 includes reservoirs at lower altitudes with higher temperatures, and located in forested areas. These reservoirs exhibit the highest $CO_2$ emissions among all categories. M-Category 0 and M-Category 2 show substantial spatial overlap with C-Category 0 and C-Category 2 and primarily consist of hydroelectric reservoirs with large mean depths and small to medium areal $CH_4$ emissions. In these groups, higher net areal $CH_4$ emissions are generally associated with larger catchment sizes.

The elucidated patterns in emission drivers and their correlation with net areal emissions highlight regions in Myanmar likely to have lower reservoir carbon footprints. They further suggest potential strategies to mitigate emissions in existing reservoirs or integrate mitigation into future investment plans. For instance, reservoirs with large catchment areas could particularly benefit from investments in wastewater treatment to reduce the influx of organic matter and nutrients, which have been shown to elevate emissions[32,33]. Meanwhile, for shallow irrigation reservoirs in the central zone, surface aeration could be effective, as their shallow depths allow for efficient oxygen penetration with low energy requirements.

**Exploring low-carbon hydropower with strategic planning**

We identified low-carbon hydropower expansion options in Myanmar using five selection criteria: GHG emissions estimated by two methods (the G-res model and global Tier 1 emission factors[4]), average and firm hydropower production, and the extent of inundated forested and agricultural land – both inputs to the G-res model.

Figure 5 a and b illustrate the Pareto-optimal trade-offs between hydropower generation targets and two emission metrics: emission intensity and total GHG emissions. Each figure compares two Pareto-optimal hydropower portfolios: (i) a built scenario ($b$), which accounts for existing infrastructure, and (ii) a hypothetical not-built scenario ($nb$), which assumes an unconstrained selection of assets. The difference between these scenarios highlights the lost opportunity for a lower-carbon hydropower pathway had GHG emissions been integrated into strategic planning from the outset. Both curves are color-coded by the total land loss metric, which represents the combined loss of agricultural and forested land. Additionally, results obtained using emissions estimated from global Tier 1 emission factor are shown in light yellow. The hydropower assets selected for three distinct generation targets, marked in Fig. 5a, b, are mapped in Fig. 5c.

We estimate the current biogenic emission intensity from hydropower at 52 $gCO_{2e}/kWh$ (G-res) and 81 $gCO_{2e}/kWh$ (EF), both exceeding the IPCC-adopted median of 24 $gCO_{2e}/kWh$[34] and slightly surpassing the 63 $gCO_{2e}/kWh$ estimate by Li et al.[35] due to several high-emission reservoirs with intensities exceeding 400 $gCO_{2e}/kWh$ (Fig. 5c, I$_b$). A lower-carbon alternative could have achieved the same hydropower generation with an average biogenic emission intensity of just 3 $gCO_{2e}/kWh$ (21 $gCO_{2e}/kWh$ including life-cycle emissions) by prioritizing run-of-river (RoR) hydropower and limiting reservoir construction to a single dam in northern

**Table 1 | Gross emission factors ($gCO_{2e}/(m^2,yr)$) by climatic zone and emission pathway, comparing global estimates[4] with values derived specifically for Myanmar**

| Climatic zone | CO2 Diffusive | | | CH4 Diffusive | | | CH4 Ebullition | | | CH4 Degassing | | |
|---|---|---|---|---|---|---|---|---|---|---|---|---|
| | Global[a] | Myanmar[c] | | Global[a] | Myanmar[c] | | Global[a] | Myanmar[c] | | Global[b] | Myanmar[c] | |
| | | All | HP | | All | HP | | All | HP | | All | HP |
| Cool temperate moist dry | 374.00 | – | – | 151.73 | – | – | 136.40 | – | – | 21.33 | – | – |
| Polar moist boreal dry moist | 344.67 | – | – | 69.52 | – | – | 24.71 | – | – | 3.28 | – | – |
| Tropical dry montane | 1081.67 | 859(20) | – | 532.77 | 387(27) | – | 800.38 | 502(59) | – | 869.13 | 1337(211) | – |
| Tropical wet moist | 1015.67 | 612(26) | 506(43) | 540.22 | 285(21) | 173(19) | 315.30 | 304(39) | 125(21) | 877.43 | 1066(218) | 834(234) |
| Warm temperate dry | 623.33 | 429(48) | 416(54) | 222.13 | 145(27) | 118(14) | 443.58 | 106(34) | 74(24) | 62.90 | 500(132) | 432(143) |
| Warm temperate moist | 535.33 | – | – | 231.44 | – | – | 201.89 | – | – | 79.55 | – | – |

Results are reported for all reservoir types (hydroelectric, multipurpose, and irrigation; All) and for hydroelectric plus multipurpose reservoirs used predominantly for hydropower generation (HP).
[a]From IPCC[17], 20 yrs integrated
[b]From Soued et al.[4], calculated using G-res tool[14]
[c]Dashes indicate missing values due to a lack of data for the corresponding climatic zones in Myanmar. Values in parentheses next to the fitted emission factors represent margins of error, calculated as half the width of the 95% confidence intervals obtained through bootstrapping.

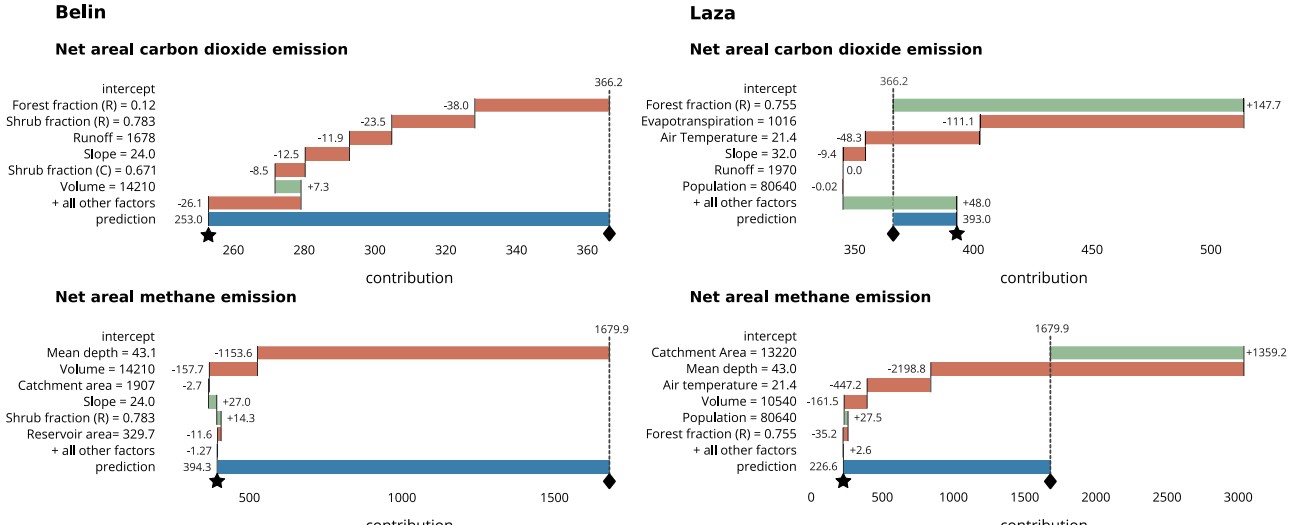

**Fig. 3 | Explanations of net areal emission predictions using additive input feature attributions and breakdown plots.** Predictions of net areal $CO_2$ and $CH_4$ emissions are explained as sums of input feature attributions. Green bars indicate positive attributions (additions to the mean prediction), red bars indicate negative attributions, and the blue bar represents the difference between the prediction and the mean predicted value across all reservoirs. The mean prediction (intercept) is represented by a diamond (◆), while the model prediction is indicated by a star (★). Input variables measured for catchments and reservoirs are labeled with (C) for catchments and (R) for reservoirs.

Irrawaddy (Fig. 5c, $I_b$ and $I_{nb}$). This strategy would have reduced annual emissions by 0.81 $MtCO_{2e}$ (G-res) and 1.26 $MtCO_{2e}$ (EF), while decreasing arable and forested land loss by 54% – equivalent to 133 $km^2$ of farmland and 106 $km^2$ of forests. For targets up to 130 TWh/year, expansion strategies rely predominantly on RoR hydropower, minimizing emissions and land loss. However, reaching 150 TWh/year (62% of total planned capacity) necessitates larger dams with greater environmental trade-offs (Fig. 5c, $III_b$ and $III_{nb}$).

The choice of emission estimation method can substantially influence asset selection, as seen for high-generation scenarios in Myanmar. Strategic planning based on G-res emissions prioritizes hydropower development along the N Mai River, whereas EF-based estimates favor projects on the Shweli River. This shift occurs despite identical values for the other four objectives, including deforestation and agricultural land loss, both derived from G-res inputs. Further optimization statistics using G-res emissions are provided in Supplementary Fig. 9. Similarities in selected asset portfolios across hydropower production targets under both GHG estimation methods are visualized in Supplementary Fig. 8 and discussed in Supplementary Results 1.2.3.

Our results show that incorporating GHG emissions into hydropower planning can help identify expansion strategies with emission intensities comparable to wind and solar energy by prioritizing low-carbon dams and RoR hydropower – though outcomes remain location-dependent. In Myanmar, we identify a production threshold beyond which emissions increase sharply, as only high-emission projects remain available for selection (Fig. 5b). Incorporating land-use metrics further refines expansion strategies by minimizing inundation, reducing the number of river barriers, and favoring less impactful upstream assets – implicitly mitigating river fragmentation. However, the choice of GHG emission estimation method as a decision criterion can substantially influence asset selection.

## Discussion

Reservoir emissions can vary widely. In Myanmar, we estimated net areal emissions from <500 $gCO_{2e}/m^2/yr$ to >5000 $gCO_{2e}/m^2/yr$ (Fig. 2), while biogenic emission intensities of hydroelectric plants range from 0.5 to 1657 $gCO_{2e}/kWh$ (Supplementary Fig. 2). This variability underscores the benefits of incorporating emissions as key criteria in reservoir planning and highlights the opportunities for carbon efficient reservoir expansion strategies. Previous studies[4,11,16,36] have used Tier 1 emission factors as estimates

of reservoir emissions. Here, we have addressed two key research questions: (1) What are the advantages of employing detailed emission models for national assessments and planning? (2) How can these models enhance the integration of emissions as a decision-making criterion?

By identifying differences of up to 3000 $gCO_{2e}/m^2/year$ and over 200% in net areal emissions between G-res and emission factors[4] (Supplementary Fig. 6), we underscore the variability in emissions that Tier 1 methodologies do not encompass. In Myanmar, EF-based total net emission estimates were 35–50% higher than those from G-res, corroborating findings from[11] and highlighting a systematic bias between both approaches. Although by calculating country-specific emission factors and net emission coefficients from G-res outputs we could reduce some of these biases, large prediction errors for individual reservoirs remained (Supplementary Tables 6, 7). These findings point to two major challenges in estimating emissions without detailed emission models: (1) the inability to resolve per-reservoir emission differences, potentially overlooking substantial variations, and (2) systematic biases that affect both gross and net anthropogenic emission predictions.

Addressing these challenges requires emission models like G-res, which account for a wide range of input variables driving emissions. However, due to their complexity, such models can introduce uncertainties that are difficult to quantify and may seem obscure to decision-makers. Sensitivity-based approaches can mitigate this issue by identifying key drivers and providing additional context for model predictions. We advocate for xAI methods due to their human-centered focus on visualization and interpretability. By analyzing model-level feature importances (Fig. 4), we demonstrated how xAI can highlight dominant emission drivers and reveal how their contributions vary across regions. Instance-level feature attribution plots (Fig. 3) are able to reveal differences between reservoirs with similar areal fluxes but distinct underlying drivers. These visual explanations enhance interpretability at both regional and individual reservoir scales, offering valuable insights for decision-making, including planning mitigation strategies.

In this work, we developed a framework for estimating reservoir emissions using G-res[14], which automates geospatial data acquisition and integrates xAI techniques to explain model outputs. Consequently, we reduced the manual effort required to process large datasets, making emission models more accessible for reservoir planning while promoting the use of higher-tier assessments over Tier 1 approximations, as recommended

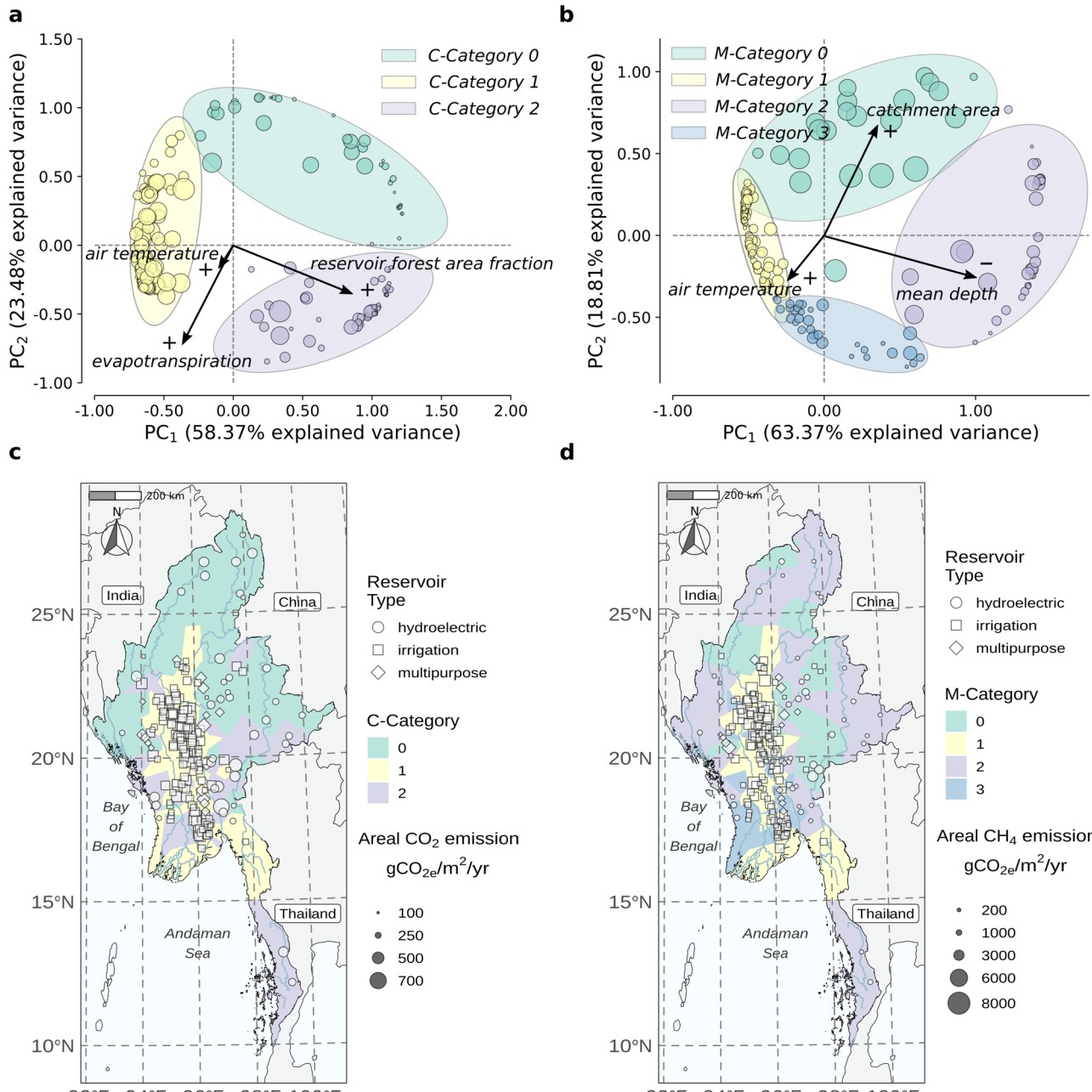

**Fig. 4 | Categorization of reservoirs with respect to $CO_2$ and $CH_4$ emission drivers. a, b,** Visualization of reservoirs in a coordinate space defined by additive input feature attributions to net areal $CO_2$ emissions (**a**) and $CH_4$ emissions (**b**) and reduced to three-dimensions using Principal Component Analysis (PCA). The first two principal directions are defined by x-axis y-axis, respectively. The third principal direction is visualized with marker size. The ellipsoids define arbitrary groups of reservoirs sharing similar emission drivers. The directions and magnitudes of the three most important input features globally are represented by arrows, which indicate the direction of increasing feature values. The signs next to the arrows indicate whether the arrow's direction corresponds to a positive ( + ) or negative ( − ) correlation with the prediction. **c, d,** Maps of reservoirs classified according to the emission driver clusters shown in **a, b**, visualized using Voronoi cells.

in the IPCC guidelines[17]. While the geospatial analysis component, supported by GeoCARET[24], is designed to be broadly applicable, the framework can be extended to incorporate other emission models and methodologies that rely on spatial data. Future developments could also expand its scope to other aquatic systems, enabling analysis of interactions between reservoirs, rivers[37,38], and surrounding landscapes. By improving transparency, this framework enhances decision-making, positioning emissions as a selection criterion alongside economic output, costs, and river health indicators.

While our results support integrating emission models into decision-making frameworks, several assumptions warrant further refinement. First of all, our framework's software can be further optimized for execution

speed, robustness, and an improved user interface for adding geospatial layers and conducting custom analyses. Second, reservoir emission modeling efforts are by no means complete and require further investigation to improve predictive power and accuracy. Reservoir emissions are governed by complex, dynamic processes that are not fully captured by static models like G-res. Factors such as water column mixing, seasonal stratification, water level fluctuations[39], drawdown zones[29], and sediment dynamics[40] substantially influence emission pathways[41]. Omitting these dynamics limits predictive power, especially under climate change and varying operating conditions. Furthermore, estimating long-term anthropogenic emissions remains challenging due to their spatial variability and non-stationarity[42],

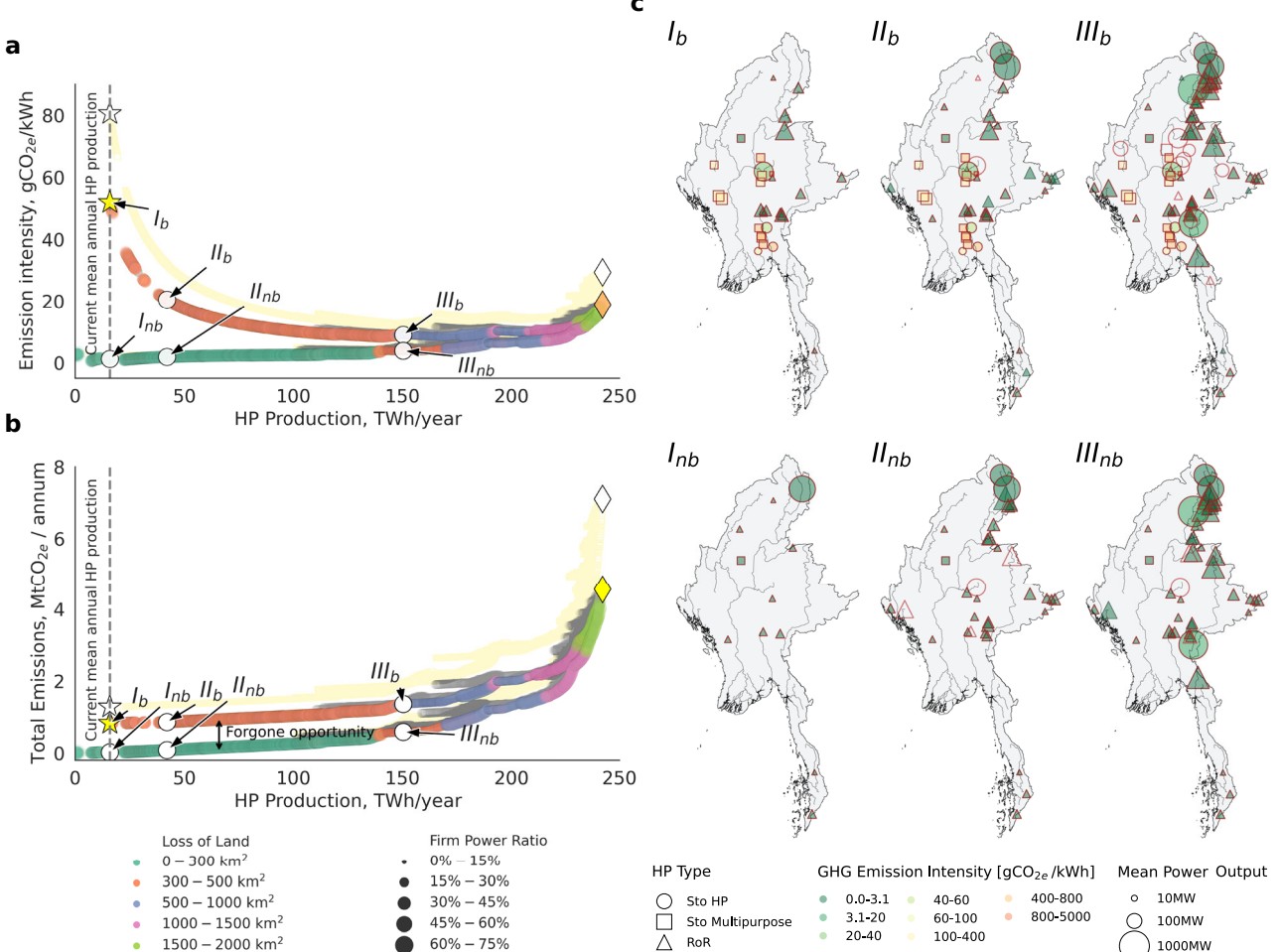

**Fig. 5 | Low-carbon hydropower planning in Myanmar.** Pareto optimal frontiers for scenarios *b* (built) and *nb* (not-built), showing trade-offs between total hydropower (HP) production and greenhouse gas (GHG) emission intensity (**a**) and total net biogenic emissions (**b**). The color-coded points represent optimal solutions obtained with emissions derived from G-res and minimizing total GHG emissions. Point color indicates associated loss of land to inundation, and curve point sizes denote the firm power ratio. Points plotted in light yellow correspond to solutions obtained with EF-based emissions. Star (★) and diamond (◆) markers indicate present-day and maximum achievable HP production, respectively. White arrow-marked points (I-III) highlight solutions selected for further analysis in **c**; where assets selected when optimizing with Tier 1 emission factors as an objective, are outlined in red.

complicating model development and calibration. Third, a better understanding of prediction accuracy requires explicit quantification of input and model uncertainties. While automated data sourcing minimizes manual effort, uncertainties in spatial datasets, such as land cover maps[43] or digital elevation models (DEMs)[44], can reduce emission prediction accuracy. Errors in DEMs may distort reservoir bathymetry, affecting estimation of littoral zones – key hotspots for $CH_4$ and $N_2O$ emissions[45,46]. In addition to input uncertainties, emission models suffer from epistemic and parametric uncertainties, with $CH_4$ ebullition being one such example. While the RE-Emission software used here supports uncertainty quantification with the Sobol method[47], this important analysis remains future work and will be essential for evaluating the reliability of emission model outputs in decision-making.

Notwithstanding these limitations, this work highlights the potential of integrating detailed emission models with automation and xAI to support reservoir-based infrastructure planning. By coupling our framework with multiobjective optimization and incorporating emissions and land loss as objectives, we could identify expansion strategies that reduce biogenic GHG emissions and promote smaller, less impactful dams in upper river reaches. In contrast, relying on Tier 1 emission estimates may result in different plans and a higher risk of selecting sites with substantially greater actual emissions. Our approach enhances prediction transparency, aligning with the "right to

explanation" principle in regulations such as the EU's GDPR[48] and broader efforts to improve model transparency[49]. The framework is readily applicable to reservoir planning at wide spatial scales, e.g., in Africa, where over 300 hydropower projects are under consideration[50], and the Amazon basin[15]. Additionally, it provides a valuable tool for detailed emissions reporting from existing reservoirs at regional and national scales.

Finally, we argue that emission models should become a routine component of reservoir assessment and planning — not as sole arbiters, but as one quantitative line of evidence among many. Emissions must be weighed alongside broader water-energy-food, environmental and social interdependencies. In our Myanmar case, many candidate reservoirs fall below an illustrative threshold of $\sim 500\,\mathrm{g\,CO_{2e}\,m^{-2}\,yr^{-1}}$ — a level that, where depth and water availability permit, yields hydropower lifecycle emissions comparable to wind and solar. Yet prioritizing only low-emission sites can impose system-wide costs (e.g., grid upgrades) and performance trade-offs, such as higher generation intermittency or increased river fragmentation. Siting and investment decisions should therefore rest on explicit multi-criteria trade-off analyses, inclusive stakeholder deliberation and clear mitigation commitments. By integrating the G-res emission model into large-scale planning and providing a foundation for future model extensions, we bring emissions from the margins to the mainstream of reservoir planning, where they stand alongside other key criteria in shaping sustainable water-energy futures.

## Methods

### Reservoir and catchment calculations

Reservoir and catchment calculations required to derive inputs for the G-res model across multiple locations were conducted using GeoCARET, an open-source Geospatial CAtchment and REservoir analysis Tool[24]. GeoCARET is interfaced with Google Earth Engine (GEE)[51], a cloud-based platform developed by Google for planetary-scale environmental data analysis, which serves as the back-end for geospatial computation and as a database of global datasets. The tool defines algorithms for performing various statistical and geometric operations necessary for deriving reservoir and catchment parameters and deploys them on GEE via its Python API. GeoCARET relies on openly accessible global datasets available via GEE as well as several additional private datasets. We made these resources accessible for the analyses conducted herein by uploading them to a dedicated GEE asset folder. The public and private assets used by GeoCARET are listed in Supplementary Table 3.

Reservoir and catchment calculations are conducted in three steps: (a) catchment delineation, (b) reservoir delineation, and (c) computation of catchment and reservoir parameters. In the first two steps, we determine the geometries of the reservoir and catchment for each dam. In the third step, we intersect multiple geospatial layers, also known as GEE assets, containing hydro-morphological and climatic data with the polygons representing the delineated reservoirs and catchments. These intersected layers are then queried to quantify various characteristic properties of the reservoirs and catchments necessary for calculating GHG emissions. The method requires a minimal input data: a dam location for analysing existing reservoirs, and a dam location plus water elevation at full supply level (FSL) for future reservoirs.

Catchment delineation is carried out using three geospatial datasets (Hydrorivers[52], Hydrosheds[53] and Hydrobasins level 12[53]; Supplementary Table 4). The dam location point is snapped to the nearest segment of the Hydrorivers network, and its sub-basin within Hydrobasins identified, denoted *S0*. The *S0* sub-basin area draining to the river-point is then calculated using Hydrosheds drainage direction data, employing a modified watershed finding algorithm of Sit et al.[54]. This delineation produces only a 'local' catchment area; the remaining "upstream catchment" is constructed by compiling all upstream sub-basins flowing to the *S0* sub-basin area, and combined with the 'local' catchment to give the complete catchment area delineation.

The method for identifying upstream sub-basins, i.e., those that flow into *S0*, is detailed below. First, we determine the highest Hydrobasin level (from 1 to 12; see ref. 53) that contains a single sub-basin fully enclosing the dam catchment, with a single source sub-basin with no inflow. This sub-basin is then designated as a region of interest (ROI) for the search area. Any level 12 Hydrobasin sub-basins outside this sub-basin is discarded. Additionally, sub-basins within the ROI that are downstream of the dam, i.e., those with Pfafstetter code numbers[55] lower than the Pfafstetter code number of *S0*, are filtered out. To refine the search further and prevent the inclusion of any downstream sub-basins by mistake, we use Pfafstetter codes to traverse sub-basins upstream within the main basin. Any sub-basins not encountered in the traversal process are discarded.

Reservoir delineation employs two distinct methodologies based on the reservoir's construction status. For planned reservoirs, delineation involves identifying a hypothetical inundated area using a hydrologically conditioned digital elevation model (DEM; Supplementary Table 4). The Full Supply Level FSL is either provided as an input or calculated by adding the dam height minus the buffer height between the FSL and the dam crest to the elevation at the base of the dam. For existing reservoirs, this method cannot be applied due to the temporal limitations (i.e., year 2000 for SRTM dataset; Supplementary Table 4) of the global DEMs, which have masked and flattened water bodies[56]. Consequently, existing reservoirs are delineated from land cover maps, as water body pixels forming a contiguous region upstream of the dam. Spatial resolutions of datasets used for reservoir and catchment delineations are detailed below. Initial reservoir and catchment delineations in raster format were vectorized before further analyses.

Reservoir and catchment parameters are generated by first interrogating a suite of geospatial layers with respect to the reservoir and catchment delineations, and subsequently applying various spatial and temporal statistical functions as required for each parameter. The computed parameters and their parent geospatial data sources are listed in Supplementary Tables 4 and 5, respectively. Parameter resolution is naturally determined by the resolution of the parent geospatial datasets, including those employed in delineating the reservoirs and catchments. Catchment delineation and analysis are conducted at a 15 Arc-second resolution, dictated by the resolution of the Hydrorivers, Hydrobasins and Hydrosheds Flow Accumulation datasets. Planned reservoirs are delineated at a finer 1 Arc-second resolution, defined by the resolution of the hydrologically conditioned DEM. For existing reservoirs, the delineation resolution is constrained to 300 meters by the land cover map.

Calculation time for each dam depends on the number of output parameters and the number and resolution of delineations defined in GeoCARET. In the 'normal' mode, where only the final outputs (impounded river section, catchment, reservoir, and catchment minus the reservoir) are exported, analyses complete in on average 4 minutes per dam. Computational and export durations naturally vary as functions of catchment area, server workload/bandwidth, but here the average values stated correspond approximately with our study median catchment size of 215 km². In the 'investigation' mode, which exports all intermediate outputs, this extends to on average 6 minutes per dam. Exporting output data separately to the user's Google Drive (GDrive) folder typically requires an additional 1-2 minutes per dam. Although the time required for calculations and data export can vary depending on factors such as the size of the delineated reservoirs and catchments, and current server workload, the average analysis time per dam is approximately 5 minutes. Further details on GeoCARET, including usage and options, are available in the technical documentation[57].

### Estimating emissions with G-res model

Emissions were calculated using RE-Emission[23,47,58], an open-source Python library for the calculation, visualization, and reporting of reservoir emissions. RE-Emission follows the G-res methodology[2], developed by the International Hydropower Association (IHA) and the UNESCO Chair in Global Environmental Change for calculating net anthropogenic emissions. It implements the G-res emission model[14,59] along with various extensions, such as $N_2O$ emissions and alternative N and P yields from land[60]. G-res is currently the most comprehensive model for reservoir emissions, incorporating a broad suite of explanatory variables that describe reservoir and catchment properties and climatic conditions unique to each reservoir's location. It provides robust empirical equations for estimating different emission pathways: diffusive $CO_2$ and $CH_4$ emissions, bubbling $CH_4$ emissions from reservoir surfaces, and $CH_4$ emissions from degassing downstream of reservoirs. These characteristics contrast with simpler models based on basic regressions and upscaling from measured data, which lack the sophistication needed to capture the complex interplay of drivers influencing emissions.

A key feature of the G-res methodology is its ability to determine the true net GHG footprint resulting from converting a river into a reservoir[2], i.e., net anthropogenic emissions, which determine the true impact of reservoirs on the atmosphere. It achieves this by calculating several GHG emission mass balances: (i) the pre-impoundment GHG footprint of the landscape, including the catchment, reservoir area, and impounded river area; (ii) reservoir emission rates for each gas and emission pathway based on various environmental factors such as climatic, geographic, edaphic, and hydrologic settings; (iii) the evolution of emission fluxes over the reservoir's lifetime; (iv) displaced emissions, or emissions that would have occurred elsewhere in the aquatic environment without the reservoir's presence; and (v) emissions related to human activity in the catchment, specifically increased emissions due to anthropogenic sources of nutrients and organic matter entering the reservoir.

Meanwhile, Tier 1 methodologies based on emission factors (e.g., refs. 10,17) estimate gross reservoir emissions, i.e., fluxes measured at the reservoir surface and in the river stretch immediately downstream. Estimating net anthropogenic emissions requires additional information on pre-impoundment emissions. As these depend on catchment characteristics and land use, Tier 1 methods cannot reliably quantify the true anthropogenic impact of reservoirs on the atmosphere (Supplementary Fig. 5).

## Country-specific Tier 1 gross and net emission factors

**Gross emissions.** Country-specific Tier 1 gross emission factors for each emission pathway in Myanmar were derived via linear regression, minimizing the error between emission factor estimates and G-res outputs. Climatic zone was used as a regressor, making the calculated emission factors a refined, Myanmar-specific adaptation of the climate-zone-based global emission factors[4]. The regression was implemented in 'scikit-learnñ', an open-source machine learning library for Python[61]. To compute 95% confidence intervals for the regression coefficients, we applied bootstrapping with 1000 resampled datasets, performing linear regression on each sample to construct a coefficient distribution. From this distribution, we derived asymptotic confidence intervals and reported margins of error as half the width of the 95% confidence intervals. The optimal emission factors were determined for three climate zones (out of the six listed by Soued et al.[4]) present in the Myanmar dataset.

**Net emissions.** Additionally, we estimated net anthropogenic emissions from Tier 1 gross emission factors using the linear equation proposed by Almeida et al.[10]. This enabled a direct comparison between net anthropogenic emissions derived from the Tier 1 methodology and those explicitly calculated using G-res.

$$F_{GHG}^{net} = \left( net_{CO_2} \times F_{CO_2} + net_{CH_4} \times F_{CH_4} \right) \times \left( 1 + R_{downstream} \right) \quad (1)$$

Here, $F_{CO_2}$ and $F_{CH_4}$ represent the gross areal $CO_2$ and $CH_4$ emissions (in $gCO_{2e}/m^2/d$), respectively. The parameters $net_{CO_2}$ and $net_{CH_4}$ define the proportion of net anthropogenic emissions in the total emissions of $CO_2$ and $CH_4$, respectively. $R_{downstream}$ specifies the ratio of downstream emissions to reservoir-surface emissions. $F_{GHG}^{net}$ is the modeled net GHG emission flux in $gCO_{2e}/m^2/d$.

The calculations were conducted using both the original parameter values[10] and calibrated values obtained through linear regression, optimizing the fit between net anthropogenic emissions estimated from Tier 1 emission factors and those derived from G-res (see Supplementary Table 6).

## Explanations with xAI and global surrogate models

The explanations of models and model outputs were computed and visualized in DALEX[31], a suite of xAI methods written in Python and R for exploring, explaining, and examining predictive models. Explainable artificial-intelligence is closely connected to the field of sensitivity analysis (SA). The similarities and differences between these two fields are discussed further in Supplementary Note 1.1.1. We applied xAI methods to explore how different features influence predictions of areal $CO_2$ and $CH_4$ emissions, as well as total emissions per a defined reservoir utility measure. For irrigation reservoirs, this measure could be emissions per crop yield, while for hydroelectric reservoirs in our case study, we used emission intensity (EI), which quantifies emissions per unit of generated energy. The explanations were provided with the help of surrogate machine-learning models, also known as metamodels, response surface models, or emulators. These surrogate models are designed to mimic the behavior of the original models. They are referred to as global because they are fitted to the entire set of modeled data and interpretable due to their architecture facilitating interpretability.

Although various explainability methods, such as iBreakDown[62], SHAP[63], or LIME[64], can be applied directly to black-box models without requiring a surrogate modeling step, we opted for the latter for several reasons: (i) it reduces computational costs, which can be substantial for feature importance calculations in models with many features, such as the G-res model (37 features, see Supplementary Table 5); (ii) it simplifies preprocessing steps like removing unwanted or correlated features, creating composite features, or applying feature scaling; and (iii) it streamlines the analysis process by eliminating the need for custom interfaces between the model and xAI algorithms, which are designed for ML models. While surrogate modeling introduces additional uncertainty that may affect metrics like explanation accuracy and faithfulness, it currently is the most flexible and replicable approach for interpreting predictions from emission and other environmental models. A more detailed discussion of surrogate modeling, including its advantages and limitations, is provided in Supplementary Note 1.1.2.

We trained two surrogate models to predict net areal emissions: one for $CO_2$ and another for $CH_4$ We chose Boosted Regression Trees (BRTs) for their high approximation accuracy, ability to handle complex nonlinear relationships and small datasets, robustness to overfitting, and high interpretability due to their hierarchical partitioning structure based on simple decision rules[65]. For implementation, we opted for LightGBM[66], an open-source, high-performance framework for gradient boosting optimized for speed and efficiency and capable of producing accurate predictive models. The methodology also supports alternative BRT implementations such as XGBoost[67] or CATBoost[68]. The surrogate model creation process is depicted in Supplementary Fig. 1. First, the models were trained using input-output data obtained from simulations using G-res. Optionally, the inputs undergo feature engineering, such as feature scaling (to enhance training speed and accuracy) feature pruning (to create parsimonious models) and introduction of new, e.g., more informative compound features. Details about the feature engineering steps undertaken in our model training are available in the accompanying source code.

The models were trained with hyperparameter tuning using scikit-learn[61] and hyperparameter optimization framework Optuna[69]. The search optimized an out-of-fold root mean squared error metric via 5-fold cross-validation with early stopping (Optuna $n_{trials}$ = 50, num_boost_round=1000, early_stop=50); the LightGBM search space included learning rate, tree complexity (num_leaves, max_depth), feature/bagging fractions and L1/L2 regularization to control model complexity. Final models were refit on the full training data using the best cross-validated parameters. Quantitatively, the surrogates attain very low relative errors and high explained variance (Supplementary Table 3; $CO_2$: RRMSE = 1.69%, RMAE = 0.70%, $R^2$ = 0.995; $CH_4$: RRMSE = 1.92%, RMAE = 1.34%, $R^2$ = 0.999), indicating strong predictive performance on held-out folds. The error diagnostic plot (Supplementary Fig. 10) indicates that absolute and relative errors remain small across the central prediction range, with a modest increase in variation at extreme predicted values but no pronounced systematic bias.

During hypertuning, we explicitly constrained model complexity (regularization, feature/bagging fractions and early stopping) and selected models by cross-validated error to promote mild generalization (i.e., to mitigate excessive overfitting) while retaining high predictive accuracy. This reflects an intentional trade-off: although further improvements in fit can be obtained in principle, strongly over-parameterized or over-fitted surrogates tend to yield less smooth and less stable post-hoc feature attributions, whereas a more parsimonious surrogate improves the reliability of subsequent xAI interpretations. Future work could further improve the balance between predictive fit and explanation stability by augmenting the effective training set with synthetic data.

Lastly, we applied xAI methods to the surrogate models to derive instance-level and model-level explanations. Instance-level explanations were computed and visualized using break-down (BD) plots[62]. BD plots provide a concise visual representation of input contributions to predictions by decomposing typically nonlinear attributions into a linear model, enabling straightforward interpretation of attributions as sums. Although not needed for our study, BD plots also detect and visualize interactions

between variables induced by the model's nonlinear nature, which causes a dependence of additive attributions on the ordering of input variables.

Model-level explanations were calculated with permutation-based feature importances[70], which gauge the impact of removing an input feature on prediction by perturbing its values. This method calculates importances through multiple permutations (e.g., 10 times to account for uncertainty related to random selection in permutations), where input values are shuffled, and measures the change in a loss function such as root-mean-square-error (RMSE) between original and permuted features. The resulting importances are ranked by magnitude, illustrating each feature's overall significance across all model predictions. The three dominant emission drivers for each gas emission were plotted with arrows in a coordinate system in defined by instance-level additive feature attributions reduced to two dimensions using PCA - see Fig. 4.

## Multiobjective hydroelectric reservoir investment planning

We conducted optimal dam selection using the multiobjective optimization algorithm for approximating Pareto frontiers on tree-structured networks of Bai et al.[71]. The algorithm offers several advantages over heuristic multiobjective evolutionary algorithms (MOEAs) commonly used in similar studies. Unlike MOEAs it provides provable assurances of approximating the Pareto frontier within an adjustable precision $\varepsilon$ which defines the maximum acceptable difference in objective values between two solutions to consider them non-dominated. Additionally, it operates in polynomial time relative to the size of the instance and $1/\varepsilon$, despite the NP-hard nature of the problem[72]. In contrast, MOEAs do not provide optimality guarantees, may require substantial computational resources and are sensitive to hyperparameter choices, potentially failing to capture entire Pareto fronts[73] or generate sparse Pareto fronts[73]. Since the algorithm effectively identifies Pareto fronts which are dense within specified approximation accuracy[10,72], the outputs can serve as candidate solutions for subsequent optimization frameworks that account for additional objectives, models, and conditions. These may include factors like climatic variability, hydrologic effects, or dynamic interactions with models from other domains, such as energy or economics.

We defined the optimization problem by assigning objective criteria values to each reservoir-based and RoR hydropower unit and establishing a directed tree that depicts the structure of connections between the hydroelectric units. In this tree structure, nodes correspond to continuous regions of the river network, while dams are modeled as directed edges connecting these nodes. These edges are oriented from downstream river sections (outlets) to upstream sections (sources). To construct the tree network, we utilized Myanmar's water resources model developed in the open-source Python Water Resources (Pywr) simulation library[74]. In the original graph of this model, dams are represented as nodes rather than edges. We transformed it into the required representation by constructing the line graph of the original graph using `networkx`[75], followed by additional custom processing steps – see Code availability.

Hydropower selection was guided by five objective criteria: total annual hydropower (HP) production and firm power (maximized), and GHG emissions, loss of agricultural land, and deforestation (minimized). These objectives represent either sums (for HP production and firm power) or means (for the remaining objectives) across all selected assets in a solution. GHG emissions were calculated with two methodologies: the G-res model[14,59] and via the emission factors from Soued et al.[4]. Total annual hydropower production and firm power for each unit were simulated using a national-scale model implemented in Pywr. The simulation was run with a historical 38-year long input time-series at a weekly time-step, for climatic forcings and river streamflows from January 1979 to December 2016. Hydropower production for each reservoir was estimated using the hydropower formula, where the head was calculated from the reservoir volume using reservoir's bathymetry and the flow was determined by the reservoir's operating rule, at each simulation time step. Firm power was estimated as the 5th percentile of the hydropower production time series. A comparison of total hydropower production and firm power, derived from

data and calculated in a water resources model, is illustrated in Supplementary Fig. 3. The empirical data were sourced from the database of dams in Myanmar shared by the Open Development Initiative[76].

The loss of agricultural land and deforestation for each reservoir were computed in GeoCARET as the areas of croplands and forests in the impounded area, respectively. These values are assumed to be zero for RoR hydropower. Reservoir emissions were calculated using RE-Emission, whilst for RoR hydropower, we assumed a constant biogenic emission intensity of 3 $gCO_{2e}$/kWh. Life-cycle emissions were calculated by adding 19 $gCO_{2e}$/kWh to the biogenic emission intensity to account for infrastructure and supply chain emissions, following Intergovernmental Panel on Climate Change (IPCC) guidelines[77]. No distinction was made between storage and RoR hydropower for infrastructure and supply chain emissions due to the wide and overlapping ranges of values reported for both[78].

## Reporting summary

Further information on research design is available in the Nature Portfolio Reporting Summary linked to this article.

## Data availability

All the data required to replicate this study are supplied together with the source code. The data can be downloaded either using a Python script available at https://github.com/tomjanus/ghg_emissions_myanmar/blob/main/download_ext_data.py or manually by following the links in https://github.com/tomjanus/ghg_emissions_myanmar/blob/main/config/data_sources.yaml. The data includes: (a) inputs required to perform the analysis, such as dam locations and full supply levels, annual hydropower production and firm power, and the time-series of hydropower production simulated in the water resources model; (b) intermediate results of computationally intensive tasks such as reservoir and catchment analysis, GHG emission estimation, training machine-learning models, and dam portfolio optimization; (c) additional data used for presenting the results. The interactive map showing the delineated reservoirs and their emissions and the topology of the water resources model, is provided at https://tomjanus.github.io/mya_emissions_map/. The results of the multiobjective hydropower selection study using emissions calculated with G-res can be viewed in a parallel axis plot published at https://tomjanus.github.io/ghg-mya-moo-results/. The outputs of gas emission calculations from RE-Emission including the final emissions report in a PDF format and all reservoir and catchment delineations, can be downloaded from FigShare using the following link: https://figshare.manchester.ac.uk/collections/PLANNING_LOW-_CARBON_RESERVOIR_INVESTMENTS_WITH_EMISSION_MODELS/7281946.

## Code availability

The software for analysing reservoirs and catchments and for estimating gas emissions is available at https://github.com/Reservoir-Research/geocaret and https://github.com/tomjanus/reemission, respectively. Both codes are under active development; for reproducibility, we froze the versions used in this study and created a `comms_earth_env` branch in each repository, containing snapshots of the code corresponding to the versions used here. Analyses reported in this study should be replicated using the code in these branches. The documentations for both software packages are provided in https://reservoir-research.github.io/geocaret/index.html and https://tomjanus.github.io/reemission/index.html, respectively. The repository of scripts in Python and R to reproduce the results and the figures can be accessed at https://github.com/tomjanus/ghg_emissions_myanmar. The repository includes the C++ source code for dam portfolio selection that has been modified from the original version[71,79] to accommodate additional objectives introduced in this study. It is available at https://github.com/tomjanus/ghg_emissions_myanmar/tree/main/moo_solver_CPAIOR.

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

## Acknowledgements

The authors would like to acknowledge the assistance provided by the Research IT Services at the University of Manchester. Special thanks to Dr. Kamilla Kopec-Harding, for her invaluable support and patience, and for developing GeoCARET. We extend our gratitude to Chris Evans for his insightful review of an early version of this manuscript and to Bogumil Ulanicki for his valuable recommendations during the later stages. This project was supported by the UK Economic and Social Research Council's Global Challenges Research Fund (GCRF) as part of the UK Research and Innovation (UKRI) Funded FutureDAMS project (ES/P011373/1). Further funding for T.J., K.K.H., and J.K. was provided by the University of Manchester to complete the GeoCARET and RE-Emission software. The work of C.B. was supported by the UK Center for Ecology and Hydrology (UKCEH).

## Author contributions

Tomasz Janus conceived the study, developed the methodology, conducted the investigation, developed the software, produced visualizations, wrote the original draft, and reviewed and edited the manuscript, Chris Barry conceived the study, developed the methodology, estimated reservoir emissions, reviewed and edited the manuscript, Shelly Win obtained the resources and reviewed the manuscript, Jaise Kuriakose conceived the study, developed the methodology, reviewed and edited the manuscript, supervised the study, and was responsible for funding acquisition.

## Competing interests

The authors declare no competing interests.
