## [Transparent Peer Review file · Communications Earth & Environment]

Planning with emission models reduces the carbon footprint of new reservoirs

Corresponding Author: Dr Tomasz Janus

Version 0:

Decision Letter:

Dear Dr Janus,

Your manuscript titled "Planning with emission models to reduce the carbon footprint of new reservoirs" has now been seen by our reviewers, whose comments appear below. In light of their advice we are delighted to say that we are happy, in principle, to publish a suitably revised version in Communications Earth & Environment.

We therefore invite you to revise your paper one last time to address the remaining concerns of our reviewers. At the same time we ask that you edit your manuscript to comply with our format requirements and to maximise the accessibility and therefore the impact of your work.

EDITORIAL REQUESTS:

****Please take care to match our formatting and policy requirements. We will check revised manuscript and return manuscripts that do not comply. Such requests will lead to delays. ****

SUBMISSION INFORMATION:

OPEN ACCESS:

Communications Earth & Environment is a fully open access journal. Articles are made freely accessible on publication. For further information about article processing charges, open access funding, and advice and support from Nature Portfolio, please visit <https://www.nature.com/commsenv/open-access>

Link Redacted

Best regards,

Somaparna Ghosh, PhD
Associate Editor,
Communications Earth & Environment
Consulting Editor,
Communications Sustainability

REVIEWERS' COMMENTS:

Reviewer #1 (Remarks to the Author):

Janus et al makes a useful contribution by integrating an automated geospatial workflow, the G-res reservoir-emission model and a tree-boosting surrogate interpreted with explainable AI. Applying this toolset to Myanmar's planned hydropower expansion, the authors show that strategic siting could reduce projected emissions while meeting most power generation targets. The study is timely and technically sound. This study region (Myanmar) is also under-represented. I believe it can be accepted after the following minor revision.

First, it's about data quality and the size of unavoidable uncertainties. The methods section explains that reservoir and catchment parameters relies on openly accessible global data, which range from 1-arc-second DEMs for planned dams to 300-m land-cover grids for existing reservoirs. It would help if the paper quantified their influence of these different resolutions, perhaps by doing the emission calculation for a subset of dams with coarser and finer inputs and reporting the resulting uncertainty spread.

Second, the surrogate model underpins all explainable-AI results, yet its fidelity to the original G-res outputs is not reported. The text outlines the breakdown and permutation-importance procedures that make the predictions interpretable, but readers need a basic fit statistic— R^2 , RMSE or mean absolute error—over the full reservoir set, together with a note on whether errors grow at very high or low emission values. Some additional details in the results or the supplement would suffice.

Finally, the paper would benefit from a brief concluding paragraph that clearly indicate an actionable rule of thumb—for instance, a recommended emission-intensity threshold or a decision tree for early-stage dam screening—so that people have a concrete takeaway.

Reviewer #2 (Remarks to the Author):

This paper uses a new, automated framework for using reservoir emissions models into GHG emissions planning and inventory development. Using Myanmar as a case study, the authors show that significant improvement in emissions estimations can be made using these tools over IPCC Tier 1 default emission factors.

While I am an expert in emissions inventories – I am not an expert in emissions from reservoirs nor machine learning. I felt this article was well explained, well organized, and had a good balance of detail and higher level explanation to be useful to readers with a variety of levels of expertise. Extensive methodological sections on xAI methods and emissions modeling provide both high level overview for those less familiar to understand the details of the methodologies and methodological detail to get into the weeds of the analysis.

I have no specific comments for the authors – after reading the article I did not feel any glaring gaps in my understanding, nor typos or hard to understand text. From reading this article, I learned a lot of xAI methods and nuances of reservoir emissions modeling and felt it provided ample insight into the decision-making process for using this tool as well as the detail behind the draw backs of using default EFs for emissions estimation and reservoir planning.

I would recommend this article for publication.

** Visit Nature Portfolio's author and referees' website at www.nature.com/authors for information about policies, services and author

benefits**

University of Manchester
Department of Mechanical, Aerospace and Civil Engineering
Manchester, United Kingdom

September 4, 2025

The Editor
Nature Communications Earth & Environment

Dear Editor,

We are grateful for the opportunity to revise and resubmit our manuscript with now a slightly modified title following Editor's suggestion "*Planning with emission models reduces the carbon footprint of new reservoirs*" to *Nature Communications Earth & Environment*. We thank both reviewers for their careful and constructive evaluations of our work. Their comments have been highly valuable in clarifying the scope of the manuscript and in improving its presentation. In what follows, we respond in detail to each of the points raised.

Response to Reviewer #1

Comment 1: On the influence of input data resolution and reporting the uncertainty spreads.

Response: We appreciate the reviewer's insightful observation that the quality and resolution of input data are central to the robustness of reservoir emission estimates. We fully agree that a systematic examination of uncertainties is indispensable if model outputs are to serve as reliable criteria for decision-making.

However, uncertainties in this domain are inherently multidimensional. They stem not only from DEM resolution, but also from land cover data, hydrological inputs, model parameterisations, and broader epistemic limitations. Assessing a single source in isolation, while feasible, would provide only a partial picture and could risk implying that uncertainty is dominated by one factor.

For this reason, in the present manuscript we explicitly acknowledge the need for comprehensive uncertainty analysis but refrain from reporting results for only one source of uncertainty. This remains an active research agenda. Our recent pre-print (<https://www.researchsquare.com/article/rs-7252618/v1>) that we now cite in the manuscript in Discussion, presents progress in this direction, and we anticipate further contributions that illuminate the impacts of various uncertainty sources on emission predictions in the wider context of calculating emission budgets and decision-making.

Comment 2: On the fidelity of the surrogate model and the need to report fit statistics.

Response: We thank the reviewer for this important suggestion. We now report standard fit metrics (R^2 , RRMSE, and RMAE) for the two surrogate models (for net areal CO₂ and CH₄ emissions) in the main text in Section 3.4 (Methods). Additionally, we provided box plots showing the distribution of errors over the entire range of predicted emission values (Supplementary Fig.10) and provided fit metric values and definitions in Supplementary Table 3. This addition ensures that the interpretability analyses presented in the paper are grounded in a clear understanding of model fidelity. We also added a note about model under- and over-fitting that are vital for understanding the model quality for explanations. In particular of the trade-off between overfitting (which improves the fit statistics at the cost of reliability of predictions) and under-fitting (that leads to surrogate models worse quality of fit but could potentially

improve interpretability in terms of identifying the relative magnitudes and ordering of most important features for individual predictions and overall). Striking the right balance between under- and over-fittings is an art and should be explored further – particularly in the context of interpretability

Comment 3: On the desirability of a clear concluding paragraph with actionable insights.

Response: We thank the reviewer for this valuable suggestion. While we did not adopt a single prescriptive rule (such as a decision tree for screening), we fully agree with the spirit of the comment and have added a brief concluding paragraph that provides a concrete takeaway. In it, we emphasize the importance of integrating emissions as a routine criterion in reservoir assessment and planning, alongside broader water-energy-food, environmental, and social interdependencies. We note an illustrative threshold of $\sim 500 \text{ g CO}_{2e} \text{ m}^{-2} \text{ yr}^{-1}$ – below which hydropower lifecycle emissions are comparable to wind and solar – yet stress that prioritizing only low-emission sites can entail system-wide trade-offs. The conclusion therefore underlines the need for explicit multi-criteria analyses, inclusive deliberation, and mitigation commitments. By mainstreaming the G-res model into large-scale planning and providing a framework for further emission model developments, we aim to make this integration practical and to ensure that emissions stand as a co-equal criterion in shaping sustainable water-energy futures.

Response to Reviewer #2

We thank the second reviewer for their generous and positive assessment of the manuscript. As no further modifications were requested, no specific changes were made in response. We are, however, grateful for their endorsement of both the methodological clarity and the broader relevance of the study.

Summary of Revisions

In summary, we have:

- Clarified the limitations of reporting partial uncertainty analyses and explained why a more comprehensive treatment is required beyond the scope of this work.
- Added surrogate model fit metrics (R^2 , RMSE, MAE) in Main in the Supplementary Information, together with notes and graphical visualization of error behaviour.
- Added a new concluding paragraph at the end of Discussion that provides a brief summary and a takeaway message.

Additionally:

- We read the manuscript and removed the remaining minor spelling mistakes.
- We added a minor change to the title, as suggested by the Editor.
- We added ORCIDs for every co-author of the manuscript.
- We slightly modified reporting emissions in Results (1.1 Reservoir emissions in Myanmar) and Table 1. We now report country-specific emission factors separately for all reservoirs and hydroelectric reservoirs. Additionally, we compare and report emission assessments nationally with and excluding degassing in irrigation reservoirs. The rationale for these small alterations is provided in the manuscript.

We provide additional PDF versions of the manuscript and supplementary materials, with all revisions relative to the original submission highlighted using `latexdiff`. Please note that

these ‘diffed’ PDFs are not flawless: some references are missing, and the version of the main manuscript excludes Table 1, as its inclusion caused compilation issues in **TeXLive**. Nevertheless, we believe these files offer a clear overview of the changes introduced during this revision.

We hope that these revisions and clarifications satisfactorily address the reviewers’ concerns. We remain grateful to the reviewers and the Editor for their constructive engagement with our work.

Sincerely,

On behalf of all co-authors
Tomasz Janus